# Novel Fluorescence-Based Methods to Determine Infarct and Scar Size in Murine Models of Reperfused Myocardial Infarction

**DOI:** 10.3390/cells13191633

**Published:** 2024-09-30

**Authors:** Ashley Duplessis, Christin Elster, Stefanie Becher, Christina Engel, Alexander Lang, Madlen Kaldirim, Christian Jung, Maria Grandoch, Malte Kelm, Susanne Pfeiler, Norbert Gerdes

**Affiliations:** 1Division of Cardiology, Pulmonology and Vascular Medicine, Medical Faculty and University Hospital, Heinrich Heine University, 40225 Düsseldorf, Germany; ashley-jane.duplessis@uni-duesseldorf.de (A.D.); christin.elster@hhu.de (C.E.); stefanie.becher@med.uni-duesseldorf.de (S.B.); christina.engel@hhu.de (C.E.); lang@hhu.de (A.L.); madlen.kaldirim@med.uni-duesseldorf.de (M.K.); christian.jung@med.uni-duesseldorf.de (C.J.); malte.kelm@med.uni-duesseldorf.de (M.K.); pfeiler@hhu.de (S.P.); 2Cardiovascular Research Institute Düsseldorf (CARID), Medical Faculty, Heinrich Heine University, 40225 Düsseldorf, Germany; maria.grandoch@uni-duesseldorf.de; 3Institute of Translational Pharmacology, Medical Faculty, Heinrich Heine University, 40225 Düsseldorf, Germany

**Keywords:** myocardial infarction, infarct size, scar size, histology, 3R

## Abstract

Determination of infarct and scar size following myocardial infarction (MI) is commonly used to evaluate the efficacy of potential cardioprotective treatments in animal models. However, histological methods to determine morphological features in the infarcted heart have barely improved since implementation while still consuming large parts of the tissue and offering little options for parallel analyses. We aim to develop a new fluorescence technology for determining infarct area and area at risk that is comparable to 2,3,5-triphenyltetrazolium chloride (TTC) staining but allows for multiple analyses on the same heart tissue. For early and late time points following MI, we compared classical histochemical approaches with fluorescence staining methods. Reperfused MI was induced in male mice, the hearts were extracted 24 h, 7-, 21-, or 28-days later and fluorescently stained by combining Hoechst and phalloidin. This approach allowed for clear visualization of the infarct area, the area at ischemic risk and the remote area not affected by MI. The combined fluorescence staining correlated with the classic TTC/Evans Blue staining 24 h after MI (r = 0.8334). In later phases (>7 d) post-MI, wheat germ agglutinin (WGA) is equally accurate as classical Sirius Red (r = 0.9752), Masson’s (r = 0.9920) and Gomori’s Trichrome (r = 0.8082) staining for determination of scar size. Additionally, feasibility to co-localize fluorescence-stained immune cells in specific regions of the infarcted myocardium was demonstrated with this protocol. In conclusion, this new procedure for determination of post-MI infarct size is not inferior to classical TTC staining, yet provides substantial benefits, including the option for unbiased software-assisted analysis while sparing ample residual tissue for additional analyses. Overall, this enhances the data quality and reduces the required animal numbers consistent with the 3R concept of animal experimentation.

## 1. Introduction

Myocardial infarction (MI) is a result of atherosclerotic cardiovascular disease and one of the leading causes of death worldwide [1]. Current research aims to identify potential therapeutic strategies to reduce cardiac damage and to preserve or increase myocardial function following MI. Since infarct size is commonly used as an indicator of cardiac damage and functional impairment after MI, histological measurement of infarct size is utilized to evaluate the efficacy of potential cardioprotective treatments in rodent models [2,3]. The most widely used method at early time points after MI is 2,3,5-triphenyltetrazolium chloride (TTC) staining that renders metabolically active tissue red while infarcted, while necrotic tissue remains white [3,4]. TTC in combination with Evans blue staining enables identification of the area at risk that is defined as tissue occluded from blood supply during MI [5]. The area at risk is an important variable that co-determines the final infarct size, as it demarks the fraction of the heart that is supplied by the obstructed or experimentally-ligated artery, thus harboring the risk to undergo cell death [5,6]. When the area at risk exceeds 20%, a close linear relationship is observed between the size of the risk area and the final infarct size [6]. Accordingly, determination of the AAR is prerequisite for a reliable estimation of the infarct size. Despite modifications and improvements to the original TTC staining protocol [7,8], the technique requires the use of whole heart tissue and precludes the possibility of performing additional co-staining for complementary analyses. Consequently, a large number of animals are required to address various research questions.

For later time points, collagen-based staining, like Sirius Red and Masson’s Trichrome staining are widely used to determine the scar size, as collagen is involved in fibrosis and scar formation following MI [9,10,11].

Next to these classical methods of staining, new fluorescence methods are available. Staining for filamentous (F)-actin, such as phalloidin and anti-F-actin antibodies, can determine infarct area by measuring the area of F-actin filaments altered and damaged during the infarct, as indicated by the absence of F-actin staining [10,12,13]. In addition, Wheat Germ Agglutinin (WGA) is known to stain glycoproteins, extracellular matrix, sialic acid, connective tissue, and collagen [14,15,16]. Emde et al. observed that WGA is suitable to determine scar tissue in sections of murine hearts harvested four weeks post MI [17].

Accurate assessment of infarct and scar area is essential for evaluating efficacy of potential treatments for MI and thus it is important to choose the most suitable method. Accordingly, we sought to assess fluorescent F-actin and extracellular matrix-based staining for measuring infarct and scar area, compare these to classical staining methods and to test their potential for combinatorial use with additional analysis methods, such as immune cell detection by immune fluorescence.

## 2. Materials and Methods

### 2.1. Animals

Male C57BL/6J mice aged 11–18 weeks were purchased from Janvier Labs (Saint-Berthevin, France) and used for experiments. All animal experiments were performed according to ARRIVE (Animal Research: Reporting of In Vivo Experiments) II guidelines and approved by LANUV (North-Rhine-Westphalia State Agency for Nature, Environment and Consumer Protection, approval number AZ 81-02.04.2020.A225) in accordance with the European Convention for the Protection of Vertebrate Animals used for Experimental and other Scientific Purposes. Mice were housed under standard laboratory conditions with a 12 h light/dark cycle and had ad libitum access to drinking water and standard chow.

### 2.2. Mouse Model

Shortly before surgery, mice were injected subcutaneously (s.c.) with buprenorphine (0.1 mg/kg body weight). After 30 min, they were briefly anesthetized with 3% isoflurane in an inhalation chamber and then intubated using a venous cannula. Mice were placed on a heated surgery table and anesthesia was maintained with isoflurane (2%) and oxygenated air (30% oxygen) at a respiratory volume of 0.2 to 0.25 mL and a respiratory rate of 140 breaths per minute. To induce MI, the left anterior descending coronary artery (LAD) was occluded for 45 min using a proline suture as previously described [18]. After 45 min, the suture was reopened to allow reperfusion of the LAD. Electrocardiogram and body temperature (37 °C) were constantly monitored. Mice received buprenorphine every 4 h (0.1 mg/kg body weight, s.c.) during the day in their drinking water during the night for 24 h or 3 days post-surgery and were monitored regularly until the end of the experiment.

### 2.3. Organ Harvesting

Mice were weighed and sacrificed 24 h, 7, 21, and 28 days following surgery by applying final anesthesia with ketamine (100 mg/kg bw) and xylazine (10 mg/kg bw) intraperitoneally. Mice were euthanized under anesthesia by drawing blood from the heart and subsequent organ removal. Hearts were perfused with 10 mL of ice-cold PBS (Merck, Darmstadt, Germany) and carefully cut out by tightly grabbing the vascular bundle, including the aorta, just above the liver and carefully cutting along the spine while pulling the bundle upwards. The removed heart was either placed into a dish filled with cold 0.9% sodium chloride solution (NaCl) for further TTC staining or stored in PBS at 4 °C for further processing.

### 2.4. TTC and Evans Blue Staining

Excess tissue was carefully removed to expose the aorta. The aorta was then trimmed just before the aortic arch, gently pulled over a nozzle and fastened with a piece of thread by tying two double knots. To re-obstruct the LAD-ligature during the operation, a silk thread was pulled through the exact place and firmly tied with a double knot. The heart was stained with 1% Evans Blue staining solution (Fluka Analytical; Munich, Germany) by coronary perfusion. Hearts were weighed, wrapped in cling film, and frozen overnight at −20 °C in preparation for sectioning and TTC staining. Frozen hearts were quickly unwrapped and cut into approx. 1 mm-thick slices and weighed individually. TTC staining solution (Merck, 1:100) was prepared in microcentrifuge tubes and warmed to 37 °C for 5 min before the tissue sections were immersed. Under constant agitation, the heart sections were stained with TTC solution for 7 min at 37 °C. Subsequently, the sections were removed from the solution, placed onto a glass slide (Engelbrecht Medizin und Labortechnik, Edermünde, Germany), and photographed using the Leica MZ6 microscope with the camera DFC450C (Leica, Wetzlar, Germany) with 1.25× objective magnification. Image analyses were performed using the DISKUS/32 software (Version 4.81.1620, Technisches Büro Hilgers, Königswinter, Germany) to measure different areas of each individual heart. As intended, Evans Blue staining solution stained all tissue that was not affected by the LAD occlusion blue, and TTC staining solution stained the viable tissue within the Evans Blue-negative area (area at risk) red. The remaining white tissue was necrotic, infarcted tissue. The workflow of the TTC staining is shown in Figure 1a.

### 2.5. Embedding and Sectioning of Organs

The hearts were fixed in 4% paraformaldehyde (PFA) (Alfa Aesar by Thermo Fisher Scientific, Schwerte, Germany) for 1 h, then dehydrated in 30% sucrose solution overnight, embedded in Tissue-Tek^®^ O.C.T. Compound (Sakura, Umkirch, Germany) and frozen at −80 °C for at least 24 h in preparation for cryo-sectioning. Hearts were sectioned (cryotome CM 3050 S, Leica) into 5 µm-thick slices according to a predefined scheme (Appendix A). From each heart, 10 layers were analyzed starting at the apex. Per layer, twenty 5 µm-thick sections were prepared and distributed onto 20 consecutive glass slides. The distance between the starting point of each layer was 250 µm (20 sections at 5 µm plus 150 µm trimming). The series of tissue sections of all 10 layers were distributed onto 3 object slides. Slide A contained 4 sections from layers 1–4 beginning at the apex, and slide B and C 3 sections from layer 5–7 and 8–10, respectively. The combination of object slides A, B and C provides 10 heart sections with a distance of 250 µm each and covering all 10 layers, thus giving an overview of the infarct area.

### 2.6. Sirius Red, Masson’s Trichrome and Gomori’s Trichrome Staining

Cryosections were defrosted for 20 min at room temperature (RT), fixed with 1% PFA for 10 min and rehydrated for 5 min in PBS. For Sirius Red staining, nuclei were stained with hematoxylin solution according to Gill II (Roth, Karlsruhe, Germany). The sections were stained with picrosirius red solution (Morphisto, Offenbach am Main, Germany) for 30 min and covered with Vecta Mount Aqueous Mounting Medium (Vector Laboratories, Newark, CA, USA). For Masson’s Trichrome staining, the slides were fixed with Bouin solution (Merck) for 1 h at 56 °C followed by a washing step with tap water. Nuclear staining was performed with hematoxylin solution according to Gill II. Slides were washed with distilled water and stained with Ponceau acid Fuchsin solution (Electron Microscopy Sciences, Hatfield, PA, USA) for 10–15 min. After washing with distilled water, tissue staining was differentiated with 1% aqueous phospho-tungstic acid solution (Electron Microscopy Sciences). The slides were immediately transferred to aniline blue solution (Electron Microscopy Sciences) for 5–10 min, followed by a washing step in distilled water. For differentiation, tissue sections were placed in 1% acetic acid solution (Merck) for 2–5 min. Slides were washed with distilled water. Sections were mounted with Vecta Mount Aqueous Mounting Medium (Vector Laboratories). For Gomori´s Trichrome staining, nuclei were stained with hematoxylin solution according to Gill ll. Sections were immersed in Gomori´s Trichrome stain (0.6% chromotrope 2R, 0.3% aniline blue, 1% acetic acid, 0.8% phospho-tungstic acid solution in distilled water) for 7 min and differentiated using 0.2% acetic acid solution. Sections were quickly dehydrated two times in 100% ethanol and mounted with Vecta Mount Aqueous Mounting Medium.

### 2.7. Phalloidin Staining to Visualize Infarct Area

Cryosections were defrosted for 20 min at RT and rehydrated in PBS for 5 min. Subsequently, slides were incubated with Flash Phalloidin™ Red 594 (Biolegend, San Diego, CA, USA, dilution 1:80) for 20 min in the dark. The remaining staining solution was washed away three times with PBS. Slides were embedded with Prolong Diamond Antifade Mountant containing 4′,6-diamidino-2-phenylindole (DAPI) (Thermo Fisher Scientific) to stain nuclei.

To confirm reliable Flash Phalloidin™-stained infarct area, the previously TTC-stained 1 mm-thick sections were fixed in 4% PFA for 1 h at RT. The tissue was prepped for cryo-sectioning through dehydration at 4 °C overnight in a 30% sucrose solution. Heart sections were embedded in Tissue-Tek^®^ O.C.T.™ Compound and frozen at −80 °C for 12 h. After sectioning, sections were stained with Flash Phalloidin™ and autofluorescence quenched for 5 min with the Vector^®^ TrueVIEW™ autofluorescence quenching kit (Vector Laboratories), then mounted with Prolong Diamond Antifade Mountant with DAPI. The workflow for phalloidin staining is displayed in Figure 1b.

### 2.8. Staining of Fresh Hearts with Hoechst Solution

To visualize tissue areas with undisrupted access to blood circulation (unaffected by LAD occlusion) with fluorescent dyes, a Hoechst staining was performed similar to the TTC- and Evans Blue staining procedure. Mice were anesthetized and their hearts removed, followed by exposition, and fastening of the aorta onto a cannula. In the identical area of the LAD ligation within the previously induced experimental MI, the LAD was tied again ex vivo using silk thread. A total of 500–800 µL of 1:1000 dilution of Hoechst (Thermo Fisher Scientific) was injected into the heart via the cannula. Hearts were then carefully pulled off the cannula without damaging the aorta and wrapped in plastic wrap (to prevent drying), placed in a 15 mL centrifuge tube protected from light and incubated in a rotator at 15 U/min for 2 h at RT. Following the incubation period, the hearts were carefully unwrapped and placed in a 30% sucrose solution for 30 min, then embedded in Tissue-Tek^®^ O.C.T.™ Compound and stored at −80 °C for 12 h. The next day, 14 µm-thick cryosections were generated and subsequently stained with Flash Phalloidin™ (dilution 1:80) to visualize the infarct area in the heart tissue, as previously described. Nuclei were stained with Draq5™ Fluorescent Probe (Thermo Fisher Scientific).

### 2.9. Wheat Germ Agglutinin (WGA) Staining for Scar Size Determination

Cryosections were defrosted for 20 min at RT and rehydrated in PBS for 5 min. Unspecific binding was blocked with blocking solution consisting of 0.1% saponin quillaja sp. (Merck), 0.5% Bovine Serum Albumin Fraction V (BSA, Roth) and 0.2% gelatin from cold water fish skin (Merck) for 1 h at RT. The slides were incubated with Fluorescein Isothiocyanate (FITC) conjugated WGA, a lectin from *Triticum vulgaris* (Merck, dilution 1:250) for 1 h at RT. Subsequently, unbound WGA solution was removed by washing in PBS. Autofluorescence quenching was performed according to the Vector^®^ TrueVIEW™ quenching kit and slides were embedded with Prolong Diamond Antifade Mountant containing DAPI.

### 2.10. Image Acquisition and Analysis Using Image J (FIJI) Software

The stained tissue sections were imaged using a DM6B fluorescence microscope (Leica) equipped with a DFC9000 fluorescence and a DCF4500 color camera. Images were obtained using LasX software (Leica) (Version 3.7.6.25997) and afterwards analyzed with FiJi (v1.53c) and the BioVoxxel Image Processing and Analysis Toolbox (BioVoxxel, Mutterstadt, Germany). Images of phalloidin-stained heart sections (24 h after MI) were analyzed using a macro that was written to measure the area of the phalloidin-positive signal per left ventricle (see Appendix A). To measure the infarct area, the polygon tool was used to manually outline the phalloidin-negative area per left ventricle. An additional macro was generated to quantify the area of the WGA signal in the tissue.

### 2.11. Statistics

Data are presented as mean ± standard deviation. Statistical analysis and graphical illustrations were performed using GraphPad Prism 8 (Graphpad Prism Inc., La Jolla, CA, USA). Data were tested for normal distribution using the Shapiro–Wilk test. Unpaired Student’s *t* test (two-tailed) was used for comparison of two groups and one-way ANOVA was used for the comparison of more than two groups. Pearson correlation was used to examine the relation between two methods. *p* < 0.05 was considered as statistically significant.

## 3. Results

### 3.1. Comparison between TTC and Phalloidin-Determined Infarct Sizes

The historically used combination of Evans Blue and the enzymatic TTC solution for analysis of experimental MI allows for the differentiation of non-ischemic area (blue), viable tissue within the ischemic area (red) and infarcted, non-viable tissue within the ischemic area (white) using transmitted light microscopy (Figure 1a and Figure 2a). For a direct comparison of TTC and our suggested approach with fluorescent dyes on the same heart tissue, TTC-stained and analyzed sections were subsequently embedded, cryo-sectioned (5 µm) and additionally stained with phalloidin (Figure 1b). The white infarcted region of the TTC staining matches visually with the area negative for phalloidin signal in the fluorescent staining approach (Figure 2a). The calculated infarct size of both staining methods showed a strong correlation, confirming the visual observation (Figure 2b,c). This demonstrates that phalloidin is equally as accurate as TTC in determining infarct size.

Additionally, we could mimic Evans Blue staining by retrograde infusion of the fluorescent nuclear dye Hoechst into freshly excised hearts. Hoechst reliably infiltrated regions not impacted by the LAD closure, staining only tissue not affected by ischemia and thus not at risk for infarction. Phalloidin exclusively binds to intact F-actin of cells, while no signal was detected after degradation or depolymerization of actin fibers as observed in cell death. We determined the area at risk as the region, following reclosure of the LAD, that is not perfused by Hoechst. Within this area, we could differentiate phalloidin-negative, infarcted tissue and regions that preserved viability during the ischemic insult and are phalloidin-positive (Figure 2d). Unlike the conventional TTC analysis, phalloidin staining allows for further staining of immune cells and other markers on the same or neighboring sections. For example, we were able to visualize the invasion of neutrophils 1 day after MI and could determine their location within the sectioned myocardium (Figure 2e).

### 3.2. WGA Stains Collagen and Is a Reliable Marker for Scar Size after Experimental MI

At later time points (e.g., day 7) following experimental MI, Gomori’s and Masson’s Trichrome, as well as Sirius Red staining, are commonly used to identify collagen fibers and visualize scar size. In addition to these classical procedures, we performed staining with WGA, which not only stains glycoproteins but also collagen [14,15,16]. We utilized 5 µm-thick, adjacent sections from the same heart 21 days after MI for all four methods. These analyses clearly showed that the same area within the heart is identified as the scar region by all four methods (Figure 3a). Quantitative analysis of the heart tissue showed no difference in scar size across the different staining methods (Figure 3b). Furthermore, WGA-determined scar sizes strongly correlated with those of the three classical staining analyses (Figure 3c; WGA vs. Sirius Red: r = 0.97, *p* = 0.0047; WGA vs. Masson’s Trichrome: r = 0.99, *p* = 0.0009; WGA vs. Gomori´s Trichrome: r = 0.81, *p* = 0.0979). Similar to the phalloidin-based analysis of infarct size in early reperfusion (24 h post MI), use of fluorescent WGA staining enabled additional staining of immune cells and other potential markers to specifically localize these to the scar, intermediate, and unaffected regions of the infarcted heart (Figure 3d). Since WGA stains collagen, it can be used to accurately measure scar size at later time points (i.e., >21 days following a MI) [17].

### 3.3. Staining with WGA Shows Reliable Advantage over Phalloidin for Measuring Scar Size at Later Time Points after Experimental MI

Phalloidin exclusively stains intact F-actin filaments, therefore the absence of phalloidin signal 24 h after MI implies loss of the filamentous actin structure observed in cardiomyocytes. However, intermediate time points are important for examining ongoing tissue repair as well as progression of scar formation indicated by increasing collagen content in the tissue. We performed co-staining of phalloidin for F-actin and WGA for collagen on sections 7- and 28-days post MI to compare infarct/scar size. WGA signal at day 7 post-MI appears diffuse and less dense compared to day 28 (Figure 4a,d). Our analyses show no significant differences between WGA-positive and phalloidin-negative signals at day 7 (Figure 4b), but there is a low correlation due to the widely dispersed and less dense WGA signal (Figure 4c). At later time points (day 28), a strong WGA signal indicated the advanced collagen production in the initial infarcted area (Figure 4d). The measured WGA-positive area was significantly larger compared to the area of altered actin filaments represented by the phalloidin-negative area (Figure 4e). Nevertheless, there is a strong correlation between both forms of staining (r = 0.8835, Figure 4f).

## 4. Discussion

In this work, we present the F-actin-staining reagent phalloidin and WGA staining as reliable and accurate methods to measure infarct area at early time points and scar size at late time points, respectively, after MI. We provide a detailed comparison with the classical methods for discrimination of infarct and scar size in a murine model of reperfused MI and demonstrate non-inferiority of the novel procedure while offering options for more reliable measurements and sparing tissue for additional analyses.

Our main goal was to show that staining with phalloidin is as reliable as TTC staining, a method that has been performed for decades and is considered the “gold standard” to analyze infarct size. Indeed, we show a strong correlation between the infarct size determined by TTC staining and the infarct size measured via fluorescent phalloidin staining 24 h after MI, corroborating the findings of Nishida et al. [10]. TTC staining is recorded using transmitted light microscopy where it is not possible to separate the different color nuances. Current techniques for automated analysis of TTC staining are limited because the intensity and saturation of staining varies between individual sections and hearts. Therefore, analysis methods rely on color perception of individual experimenters, which is biased and widely varied. This subjective perception is not an issue when analyzing fluorescence signal of phalloidin-stained sections, because color channels can be separated, and the analysis is binary: signal or no signal.

Phalloidin is not only as reliable as TTC, but it also produces more accurate results since the borders of tissue regions are better defined, allowing for software-assisted, semi-automatic analysis and the ability to generate uniform and similar results across experimenters and labs. This provides a clear advantage over the TTC analysis. However, the preparation of cardiac tissue for staining with phalloidin is more time-consuming as 5 µm-thick sections are generated with a cryotome. TTC staining requires at least 1 mm-thick sections to achieve the required color contrast [5]. Depending on the laboratory, these sections are made either with a tissue cutter or manually with a scalpel. The thickness of the cuts made manually with a scalpel or with a tissue cutter are never as defined and uniform as semi-automated cuts with a cryotome. Yet the main advantage of preparing only 5 µm-thick cryosections from different levels of the whole heart is the possibility of addressing additional research questions using one heart. This allows for performing, of a wide range of methods of co-staining, (e.g., immune cells or other proteins of interest) using one animal heart. Thus, the number of animals and experiments required is reduced, supporting the general effort to refine analyses and reduce animal experimentation. We have not been able to perform immune cell staining on TTC sections due to the thickness and broad emission spectra of the enzymatically developed red color.

So far, a unique benefit of TTC staining over other methods is the visualization of area at risk, meaning ischemic tissue that is devoid from blood and oxygen supply during the surgical obstruction of the LAD [5]. In our study, we were able to establish a novel methodology to visualize tissue at risk by replacing Evans Blue staining with the fluorescent Hoechst dye and combining it with phalloidin staining. Therefore, phalloidin in combination with Hoechst staining can fully replace TTC staining and even offers three benefits: (1) uniform sectioning procedure to enable reproducible results, (2) the ability to perform co-staining, thereby reducing animal number and required material while, simultaneously allowing localization of cells and proteins of interest within specific infarct regions, (3) fast, reproducible and objective analysis using software-assisted, semi-automatic analysis.

In addition, phalloidin staining alone can be used for later time points than 24 h. The critical issue with the staining procedures employing both Evans Blue, but also Hoechst in our study, is the need to re-apply the suture at the same location of the LAD as during previous MI surgery, thereby accurately determining the “real” area at risk. This can be challenging due to poor visibility of the ligation site and depth of the suture and requires an experienced and well-trained animal surgeon. Inaccurate replacement of the suture influences the size of the area at risk, resulting in a different ratio of infarct size to area at risk.

At later points after MI, scar formation is prevalent, driven by an increased release of collagen and other matrix proteins by activated fibroblasts within the damaged tissue area [19,20]. Increased scar size, often considered an indicator of poor myocardial function [21,22], is classically determined by collagen-specific staining methods (e.g., Sirius Red, Masson’s Trichrome and Gomori’s Trichrome). As shown earlier by Emde et al., WGA also binds to collagen and is comparable with Sirius Red, although it does not allow differentiation of collagen subtypes [17]. We confirm and extend these results by analyzing an additional time point (day 21 post MI) as well as additional comparisons with Masson and Gomori’s Trichrome staining. Our data show a clear correlation of the measured collagen content with the detected WGA-positive signal. All classical histochemical techniques are based on acid dye and the resulting intensive staining represents a challenge for simultaneous IHC co-staining. WGA is as reliable as the classical methods and provides the beneficial possibility of multicolor fluorescent co-staining [17]. While the initial tissue preparation of the heart for these methods of staining is the same, WGA is a simpler and faster approach to staining collagenous tissue. All classical staining, as well as WGA staining, can be analyzed using an automated measuring tool, resulting in comparable and reliable results.

We also investigated which method is best suited for measuring infarct/scar size during time points between 24 h and 28 d. It should be noted that the mRNA levels of collagen types I and III begin to increase 48 h after MI [23]. Therefore, a reliable determination of the infarct area based on collagen staining is only possible >72 h after MI [24,25]. Next to WGA, phalloidin can also be utilized at later time points to define areas affected by tissue alterations after MI, as phalloidin only binds to intact filamentous actin. We showed that the WGA-positive area is almost double the area lacking intact F-actin filaments and presumably cardiomyocytes. This is likely due to invading fibroblasts and their increased collagen release within the damaged myocardium over time [25]. A physiological reason for this effect may be to anchor the scar to the unaffected healthy tissue. It is also possible that connective tissue and glycoproteins stained by WGA minimally factor into the measurement of the WGA-positive signal. Based on our results, we recommend the use of F-actin staining with phalloidin to determine affected tissue up to 7 days post MI, while collagen-based staining by WGA is preferable to visualize scar size at time points >7 days.

In summary, we have shown that histological analysis of infarct and scar size can be equally accurately determined using fluorescence or histochemical methods. Nevertheless, we would, like to point out the advantages of fluorescent staining for visualizing infarct or scar areas in the experimental MI, as this method offers the possibility of easily separating emission spectra and semi-automated quantification, thus significantly reducing the susceptibility to human error due to subjective observation. With the combination of phalloidin and Hoechst staining, we established an additional method for clear definition of the infarct region, area at risk, and unaffected tissue after MI. This approach leads to a significant reduction in animal numbers due to the possibility of various co-stainings and the generation of larger sample numbers using only one animal. The effective use of a single heart to answer a variety of research questions is a major advantage over traditional TTC- and Evans Blue-based staining procedures. Table 1 summarizes the most relevant aspects of our data and discussion, which may be utilized as a guideline by other experimenters.

## Figures and Tables

**Figure 1 cells-13-01633-f001:**
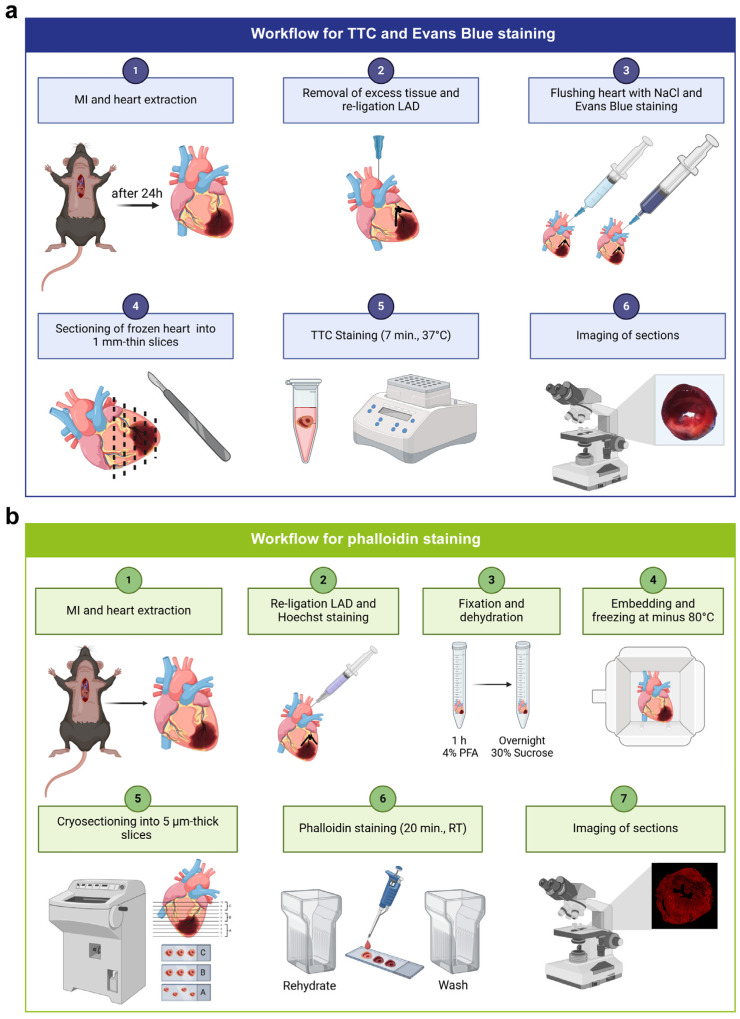
**Workflows for infarct analysis procedures**. (**a**) Workflow for TTC and Evans Blue staining procedure. After functional anesthesia, the heart is excised and fat and surrounding tissue are removed. The suture that was placed during the surgically induced MI is replaced with a silk suture to prevent loosening during the staining procedure. The heart is hung onto a cannula by the aorta and rinsed with NaCl before being stained with Evans Blue solution. Frozen tissue sections (approx. 1 mm) are cut from the apex to the base of the heart and stained with TTC solution. After staining, the sections are imaged by light microscopy and analyzed. (**b**) Workflow for phalloidin staining procedure. After functional anesthesia, the heart is excised and fat and surrounding tissue are removed. The heart is rinsed with ice-cold PBS and fixed 1 h in 4% PFA. The suture that was placed during the surgically induced MI is replaced with a silk suture to prevent loosening during the staining procedure. The heart is stained with Hoechst to identify the area at risk. Following a dehydration step in sucrose solution, tissue is embedded in Tissue-Tek^®^ O.C.T.™ Compound and deep frozen at −80 °C. Cryo-sectioning is performed, and the tissue sections are stained with phalloidin. After staining, the sections are imaged by fluorescence microscopy and analyzed. Created in BioRender. Elster, C. BioRender.com/c54f420 (2023) and BioRender.com/m01f052 (2024).

**Figure 2 cells-13-01633-f002:**
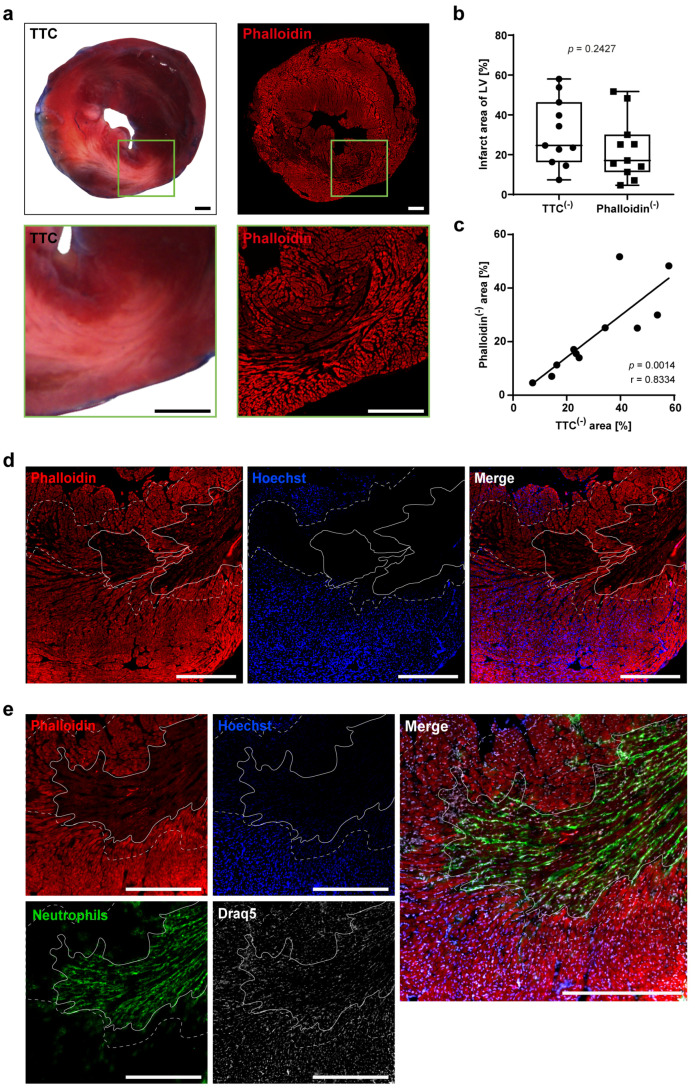
**Comparable infarct sizes determined by TTC- and phalloidin staining 24 h after experimental MI.** (**a**) The same section of an infarcted heart 24 h after MI was stained with TTC (left) and with phalloidin (right). Absence of phalloidin staining indicated infarct area and corresponds to the white area in TTC staining. The enlarged image section is marked by a green field. (**b**) Boxplot displaying infarct size per left ventricle (LV) determined by TTC- and phalloidin staining (*n* = 11; unpaired *t* test, *p* = 0.2427). (**c**) Pearson correlation of infarct size between TTC and phalloidin staining (*n* = 11; *p* = 0.0014). (**d**) Staining of Hoechst and phalloidin shows the infarct (solid line, phalloidin^(−)^; Hoechst^(−)^) and viable tissue within the area at risk (between closed and dotted line phalloidin^(+)^;Hoechst^(−)^). Double positive signal reflects viable tissue outside the area at risk not affected by ischemia. (**e**) Co-staining of infarct area with immune cells. Neutrophils (Ly6G^(+)^, green) within the infarct zone (phalloidin^(−)^;Hoechst^(−)^). Nuclei were stained with Draq5™. Scale bar 500 µm.

**Figure 3 cells-13-01633-f003:**
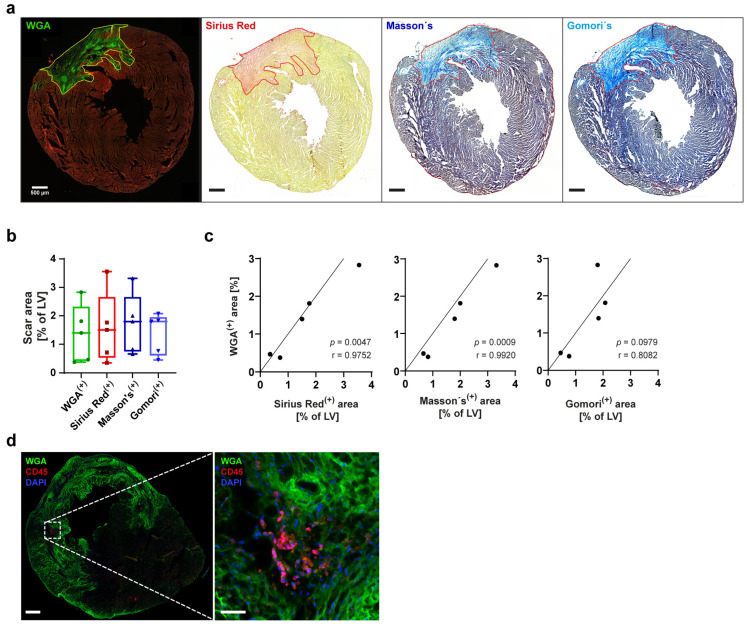
**WGA staining reliably determined scar size following MI.** (**a**) Consecutive infarcted heart sections 21 d after MI were stained with WGA, Sirius Red, Masson’s Trichrome or Gomori’s Trichrome. WGA-positive area is identical to scar tissue detected in classical staining. (**b**) Statistical analysis showed no difference in scar size between the different methods of staining (*n* = 5 for each staining, one-way ANOVA with multiple comparisons, *p* = 0.9413). (**c**) WGA scar size shows a strong correlation with Sirius Red, Masson’s and Gomori’s Trichrome scar size respectively. (*n* = 5) (**d**) Immunofluorescent co-staining of leukocytes (CD45, red) within scar region (WGA positive, green) of infarcted heart (d28). Nuclei were stained with DAPI. Scale bar 500 µm. Scale bar close-up 50 µm.

**Figure 4 cells-13-01633-f004:**
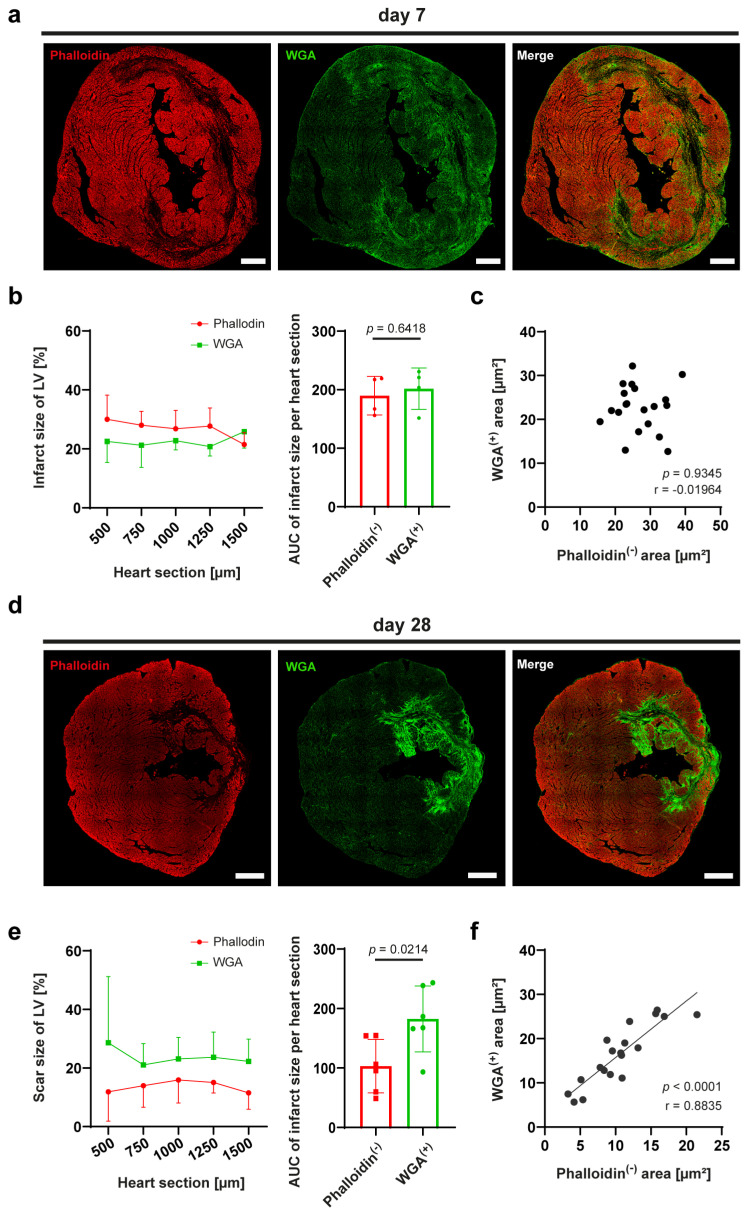
**WGA staining is more suitable than phalloidin for measuring scar size at late time points after MI.** Mice underwent reperfused MI and were euthanized 7 or 28 days later. (**a**) Localization of phalloidin (red) and WGA (green) shows no complete overlap of WGA-positive and phalloidin-negative area on day 7 after MI. (**b**) At intermediate time points (day 7) after MI, the phalloidin-negative area (infarct area) is larger than the WGA-positive area (scar area). However, this difference is not significant, as determined by comparison of area under the curve (AUC) with an unpaired *t*-test (*n* = 4; *p* = 0.6418). (**c**) Pearson correlation of phalloidin-negative and WGA-positive signal in individual mice confirms a low overlap (*n* = 20; *p* = 0.9345). (**d**) Localization of WGA (green) and phalloidin (red) on day 28 after MI shows distinct overlap of WGA positive (scar) and phalloidin-negative (infarct) area, with scar tissue permeating deeper into the healthy tissue. (**e**) At late time points (day 28) after MI the WGA-positive area (scar area) is significantly larger than the phalloidin-negative area (infarct area). Significance was determined by calculating AUC and running an unpaired *t*-test (*n* = 6; *p* = 0.0214). (**f**) Pearson correlation of phalloidin-negative and WGA positive signal shows a strong overlap of signals (*n* = 19; *p* < 0.0001). Scale bar refers to 500 µm in all panels.

**Table 1 cells-13-01633-t001:** Comparison of the different staining methods to determine infarct area, area at risk or scar area.

	TTC + Evans Blue	Phalloidin + Hoechst	Sirius Red	Gomori’s Trichrome	Masson’s Trichrome	WGA
Determination of infarct area	** ✓ **	** ✓ **	** ✕ **	** ✕ **	** ✕ **	** ✕ **
Determination of area at risk	** ✓ **	** ✓ **	** ✕ **	** ✕ **	** ✕ **	** ✕ **
Determination of scar size	** ✕ **	** ✕ **	** ✓ **	** ✓ **	** ✓ **	** ✓ **
Time required for staining	1–2 h	2–3 days	1–2 h	1–2 h	1–2 h	1.5 h
Co-staining	** ✕ **	** ✓ **	-	-	-	** ✓ **
Uniform/reproducible procedure	(**✓**)	** ✓ **	** ✓ **	** ✓ **	** ✓ **	** ✓ **
Semi-automated analysis	** ✕ **	** ✓ **	** ✓ **	** ✓ **	** ✓ **	** ✓ **

Symbols and abbreviations used: **✓**/(**✓**) parameters that can be determined fully/(partly) with the respective technique; **✕ ** parameters that cannot be measured with the technique. The dash indicates analyses that were not tested in this paper. TTC 2,3,5-triphenyltetrazolium chloride; WGA wheat germ agglutinin.

## Data Availability

The original contributions presented in the study are included in the article and Appendix A, further inquiries can be directed to the corresponding author.

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
