# Peer review of "Novel Fluorescence-Based Methods to Determine Infarct and Scar Size in Murine Models of Reperfused Myocardial Infarction"

_cells, 2024, doi:10.3390/cells13191633_

Round 1

Reviewer 1 Report

Comments and Suggestions for Authors

The authors present an elegant observational study examining whether fluorescence-based approaches are non-inferior to ‘traditional’ approaches to measuring infarction, post-MI.

Strengths are the inclusion of multiple time points and methods.

This is generally a well written and presented manuscript.

Indeed, I have no critique or concerns regarding the scientific approach, analysis or interpretation.

Minor Comments

Line 42 &43 - As noted in ref #6 (Feiring et al 1987), the size of the of the AAR is an important variable in regards to the size of the infarct. If AAR is >20%, there is a linear relationship between AAR size (volume) and final infarct size. This important relationship is somewhat unclear or lost in the current statement. It would be highly valuable to emphasize this critical relationship, given that we see more and more studies reporting infarct size in the absence of any measure of AAR.

Line 77 – “Hoechst reliably infiltrated”?

Line 131 - What is the likelihood of the original suture loosening during staining vs the replacement suture not following the same path (entry, exit and depth)? Acknowledging it is noted elsewhere in the manuscript that caution must be applied here.

Line 154 – ‘ordinary’ could be removed.

Lines 175-177 appear to be text from ‘instructions for authors’ and should be removed.

Line 221 – “critical issue with the staining procedures involving Evans Blue but also Hoechst..”?

Line 306 – “Mice died” should be replaced with “Mice were euthanised…”

I found the section, lines 340-345, confusing. 10 layers with 20 slides per layer is the 200 slides, yet the 10 layers were distributed across 3 slides? Is the 20 slides per layer referring to the n=20 (line 166)?

Author Response

The reviewers comments are given in plain text and black.

The authors responses are given in bold:

Response to reviewer 1:

Reviewer 1: Comments and Suggestions for Authors

The authors present an elegant observational study examining whether fluorescence-based approaches are non-inferior to ‘traditional’ approaches to measuring infarction, post-MI.

Strengths are the inclusion of multiple time points and methods.

This is generally a well written and presented manuscript.

Indeed, I have no critique or concerns regarding the scientific approach, analysis or interpretation.

We thank the reviewer for the positive feedback and valuable comments.

Minor Comments

Line 42 &43 - As noted in ref #6 (Feiring et al 1987), the size of the of the AAR is an important variable in regards to the size of the infarct. If AAR is >20%, there is a linear relationship between AAR size (volume) and final infarct size. This important relationship is somewhat unclear or lost in the current statement. It would be highly valuable to emphasize this critical relationship, given that we see more and more studies reporting infarct size in the absence of any measure of AAR.

We thank the reviewer for raising this important point. We have modified the paragraph adding the suggestion of the reviewer and added additional references highlighting the important relationship between AAR and final infarct volume. Please see now line 48ff.

Line 77 – “Hoechst reliably infiltrated”?

We corrected the text. Now please see line 90.

Line 131 - What is the likelihood of the original suture loosening during staining vs the replacement suture not following the same path (entry, exit and depth)? Acknowledging it is noted elsewhere in the manuscript that caution must be applied here.

We thank the reviewer for bringing up this technical detail. We have supplemented the text in the Methods section. The infarct model shown here is based on the reopening of the ligature initiating full reperfusion paralleling most clinical presentations in the cath lab where full reperfusion of obstructed coronary arteries is aimed for. It is important to note that the original knot for the ligation of the LAD was made with a Prolene suture, a material that allows the knot to be released after the ischemia phase (45 minutes) to reperfuse the tissue. This suture is removed after the operation to avoid infection or impairment of cardiac function. The suture material used for the staining approach is silk; loosening of the knot is virtually impossible due to the nature of the fibers. Nonetheless, unintended Evans Blue reperfusion is controlled for visually using the stereomicroscope.

The replacement suture is placed in the same position as the ligature during the operation. At this point, inexperienced surgeons may be prone to a certain degree of error. In our experiments, the surgery and the placement of the replacement suture are carried out by the same highly-experiences animal surgeon.

We have further highlighted this now in the Discussion section, line 239ff.

Line 154 – ‘ordinary’ could be removed.

We omitted the word, now line 170.

Lines 175-177 appear to be text from ‘instructions for authors’ and should be removed.

The reviewer is absolutely right, we apologize for this mishap and corrected the text accordingly (now line 191ff).

Line 221 – “critical issue with the staining procedures involving Evans Blue but also Hoechst..”?

We corrected the wording in the text (line 237).

Line 306 – “Mice died” should be replaced with “Mice were euthanised…”

We changed wording in the text (line 324).

I found the section, lines 340-345, confusing. 10 layers with 20 slides per layer is the 200 slides, yet the 10 layers were distributed across 3 slides?

We apologize for the confusion caused by the previous description. We now changed the wording regarding the slide preparation in the text to make it easier to comprehend. However, the procedure is probably still easiest to understand in the context of supplementary figure 1. Please see the modified text in the Methods section (line 358 ff).

Is the 20 slides per layer referring to the n=20 (line 166)?

We apologize for causing this misunderstanding. The n-number refers to 20 individual animals of which only slide B1 was used, which by our experience displays the most pronounced infarct area. We have now stated in the figure legend that the replicates refer to individual mice (line 182).

Reviewer 2 Report

Comments and Suggestions for Authors
  • The abstract lacks some details on the methodology and specific results. Refine the abstract to include  the hypothesis, key results (including any statistical significance), and the study’s main conclusion. 
  • Consider adding in the introduction more recent references to ensure that the study is up to dated within the latest developments in the field.
  • Break down the results into subsections and add more context to the figures.
  • Expand the discussion to include alternative interpretations and comparisons to existing literature.

Author Response

The reviewers comments are given in plain text and black.

The authors responses are given in bold:

Response to reviewer 2:

Reviewer 2: Comments and Suggestions for Authors

The abstract lacks some details on the methodology and specific results. Refine the abstract to include the hypothesis, key results (including any statistical significance), and the study’s main conclusion. 

We thank the reviewer for the overall insightful and valuable comments.

We have changed the wording of the abstract to present the content and major improvement of the infarct determination of the present study in more detail. In addition, we have added correlation coefficients (Line 15ff).

Consider adding in the introduction more recent references to ensure that the study is up to dated within the latest developments in the field.

We thank the reviewer for this valuable suggestion. We have expanded the introduction to include more information from currently available literature. A repeated search on existing studies that investigate infarct size quantification showed that the original histological protocols have been optimized and modified over time. However, in our view methodological developments are still not sufficient to address different questions on one and the same tissue. The literature on infarct determination is mostly focused on correct quantification of the infarct, the area at risk, or the remote area. (doi 10.1152/ajpheart.00836.2009). To date, no alternative methods for quantifying infarct area with the possibility of further tissue processing such as co-stainings have been published. The changes in the text are found in line 48ff.

Break down the results into subsections and add more context to the figures.

The results section has been adapted and now reflects the content of the figures in its chapters. See line 121ff.

Expand the discussion to include alternative interpretations and comparisons to existing literature.

We thank the reviewer for her/his suggestion. However, we are a bit unsure how to implement the requested alternative interpretations and comparisons with existing literature. To date, no analysis of an infarct region using fluorescence-based staining methods has been described. For the first time, we describe an alternative to the existing methods, showing the strong correlation between the widely used histological stains such as TTC, Sirius red, etc. with the data obtained from the newly described technique. Here, we could demonstrate that this method generates reliable, reproducible data and also opens up the possibility for additional analyses such as co-staining on the identical tissue section. Consequently, we discuss our results in the context of the current literature (see line 192ff).

Round 2

Reviewer 2 Report

Comments and Suggestions for Authors

The authors answered my questions.